# Infections of Venetoclax-Based Chemotherapy in Acute Myeloid Leukemia: Rationale for Proper Antimicrobial Prophylaxis

**DOI:** 10.3390/cancers13246285

**Published:** 2021-12-14

**Authors:** Raeseok Lee, Sung-Yeon Cho, Dong-Gun Lee, Hyeah Choi, Silvia Park, Byung-Sik Cho, Yoo-Jin Kim, Hee-Je Kim

**Affiliations:** 1Catholic Hematology Hospital, The Catholic University of Korea, Seoul 06591, Korea; misozium03@catholic.ac.kr (R.L.); cho.sy@catholic.ac.kr (S.-Y.C.); hyeah.choi@catholic.ac.kr (H.C.); silvia.park@catholic.ac.kr (S.P.); cbscho@catholic.ac.kr (B.-S.C.); yoojink@catholic.ac.kr (Y.-J.K.); cumckim@catholic.ac.kr (H.-J.K.); 2Department of Internal Medicine, Division of Infectious Diseases, College of Medicine, The Catholic University of Korea, Seoul 06591, Korea; 3Department of Hematology, Department of Internal Medicine, College of Medicine, The Catholic University of Korea, Seoul 06591, Korea

**Keywords:** venetoclax, antibiotic prophylaxis, acute myeloid leukemia, invasive fungal infections, bacteremia

## Abstract

**Simple Summary:**

Venetoclax (VEN)-based combination chemotherapy has been a promising option for acute myeloid leukemia (AML) treatment. However, the risk of infections and strategies of prophylaxis are not yet established. This study aimed to evaluate the severe infectious complications of VEN-based chemotherapy and to clarify the evidence for antimicrobial prophylaxis. The incidence of invasive fungal infections (IFIs) and bloodstream infections (BSIs) was 6.6/100 cycles and 12.7/100 cycles respectively. Secondary and therapy-related AML was an independent risk factor for IFIs (odds ratio, 3.859; 95% confidence interval, 1.344–11.048, *p* = 0.012). Patients with IFIs showed significantly poorer outcomes, but there was no statistically significant difference in patients with BSIs. Mold-active antifungal agents as prophylaxis are generally recommended in high-risk patients with AML who are treated with VEN-based combination chemotherapy.

**Abstract:**

Although venetoclax (VEN)-based combination chemotherapy in patients with acute myeloid leukemia (AML) results in prolonged and profound neutropenia, data regarding infectious complications and antimicrobial prophylaxis are lacking. We investigated the infectious complications in 122 adult patients with AML under the same standard of care for prevention. The prophylaxis protocol was fluconazole 400 mg/d without antibacterial agents. The incidence of proven or probable invasive fungal infections (IFIs) was 6.6/100 cycles, and 22 patients (18.0%) were diagnosed (median, second cycle; interquartile range, 1–2). All IFIs were caused by *Aspergillus* and significantly influenced the overall mortality (odds ratio (OR), 2.737; 95% confidence interval (CI), 1.051–7.128; *p* = 0.034). In the multivariate analysis, secondary or therapy-related AML was an independent risk factor for IFIs (OR, 3.859; 95% CI, 1.344–11.048, *p* = 0.012). A total of 39 bloodstream infection (BSIs) episodes occurred in 35 patients (28.7%), with an incidence of 12.7/100 cycles. High-dose steroid administration within 90 days was associated with the occurrence of BSIs (OR, 7.474; 95% CI; 1.661–3.631, *p* = 0.008), although BSIs themselves did not have an impact on the outcomes. Our findings suggest evidence for the need for mold-active antifungal agents as antifungal prophylaxis, rather than fluconazole, especially in patients with secondary or therapy-related AML.

## 1. Introduction

Venetoclax (VEN) is a potent inhibitor of anti-apoptotic B-cell leukemia/lymphoma-2 (BCL-2) which is overexpressed in leukemia stem cells [1,2] Clinical data have demonstrated that VEN-based combination chemotherapy with hypomethylating agents (HMAs) or low-dose cytarabine (LDAC) leads to more desirable outcomes than HMAs or LDAC alone in patients with acute myeloid leukemia (AML) [3,4,5,6]. Recently, a VEN-based combination with HMAs or LDAC has been approved for the treatment of newly diagnosed (ND) elderly AML patients and has emerged as a promising therapeutic option for relapsed/refractory (R/R) AML as well [3,7,8,9].

For patients with AML, antimicrobial prophylaxis, including use of antibacterial and antifungal agents, is recommended in order to reduce the risk of infections, including invasive fungal infections (IFIs) and bloodstream infections (BSIs), due to prolonged and profound neutropenia during intensive chemotherapy [10,11,12]. Although VEN-based combination chemotherapy is known to cause prolonged and profound neutropenia frequently, data regarding antimicrobial prophylaxis for VEN-based combination chemotherapy in patients with AML are lacking [13,14]. VEN is known to be associated with a low risk of infection when administered as a single agent, and studies have reported that the risk of severe infection in VEN-based combination chemotherapy is not significantly higher than that of monotherapy using HMAs or LDAC [3,6,15]. Rather, recent retrospective cohort studies have reported a significant risk of infection in patients undergoing VEN-based combination chemotherapy, with IFIs incidence rates over 6.9–12.6% [14,16]. However, in previous studies, a various spectrum of antifungal and antibacterial agents has been used, making it difficult to accurately evaluate the risk of severe infections. Although posaconazole is the current standard of the prophylactic agent in patients with AML undergoing remission induction intensive chemotherapy, its efficacy in biological or targeted agents for patients with AML was not thoroughly evaluated [11].

This study aimed to evaluate the incidence and risk factors of severe infectious complications, such as IFIs and BSIs, during VEN-based combination chemotherapy when used in ND and R/R settings in patients with AML under the uniform standard care for prophylaxis. We also examined the effectiveness and limitations of fluconazole prophylaxis.

## 2. Materials and Methods

### 2.1. Study Population and Study Design

We retrospectively reviewed data for all adult patients (≥18 years old) with AML treated with VEN-based combination chemotherapy between January 2020 and March 2021. The outcomes and responses were evaluated until May 2021 at the Catholic Hematology Hospital, which performs over 500 hematopoietic stem cell transplantations (HSCT) annually. IFIs and BSIs were analyzed from the initiation of VEN-based chemotherapy until the earliest of the following: (1) starting date of conditioning chemotherapy for the receipt of HSCT; (2) 7 days after switching to other chemotherapeutic regimens; (3) 30 days after discontinuing VEN-based combination chemotherapy; (4) death [14]. This study was approved by the Institutional Review Board of Seoul St. Mary’s Hospital with a waiver of informed consent, due to the retrospective study design (KC21RASI0658).

### 2.2. Definitions

Proven, probable, or possible IFIs (as per the European Organization for Research and Treatment of Cancer/Mycoses Study Group) were recorded, and breakthrough IFIs were defined based on the consensus definition in the European Confederation of Medical Mycology criteria [17,18]. We used the World Health Organization 2016 classification and the 2017 European Leukemia Net risk stratification system to classify the AML type and risk [19,20]. The response group included patients who had achieved complete recovery (CR), those who achieved CR with an incomplete hematologic recovery (CRi), and those who achieved a morphologic leukemia-free state (MLFS) after VEN-based combination chemotherapy according to the International Working Group criteria [21].

### 2.3. Dosage of VEN-Based Combination Chemotherapy and Antimicrobial Prophylaxis

Decitabine (20 mg/m^2^ intravenously daily for 5 days), azacitidine (75 mg/m^2^ intravenously daily for 7 days), or low-dose cytarabine (20 mg/m^2^ subcutaneously daily for 10 days) was combined with VEN, with the VEN dose starting at 100 mg on day 1 and increasing to 400 mg or to 600 mg (only for low-dose cytarabine), as previously described [3,5,6]. The VEN dose was reduced by 75% when combined with a strong inhibitor of CYP3A4, such as posaconazole, while it was reduced by 50% when combined with a moderate inhibitor, such as fluconazole [13,22]. It was intended to use fluconazole 400 mg/day for antifungal prophylaxis during the neutropenic period in all patients, regardless of their type of AML, underlying disease status, and response. Antibacterial prophylaxis was not administered at our institution.

### 2.4. Statistical Analysis

The characteristics of the study population were summarized using numbers and percentages for categorical variables and median values with interquartile range (IQR) or ranges for continuous variables. Categorical variables were analyzed using the χ2-test or Fisher’s exact test. Continuous variables were examined using the Wilcoxon signed-rank test for non-parametric methods. A logistic regression model was performed to assess the independent risk factors for IFIs or BSIs, and assumptions for autocorrelation and linear regression were checked. For survival analysis, only the first episode of IFIs or BSIs occurring during the study period was used. The Kaplan-Meier method with the log-rank test was used to determine overall survival.

## 3. Results

### 3.1. Baseline Characteristics of Study Patients

A total of 122 adult patients who had AML, with a median age of 61 years (interquartile range (IQR), 47–70), were enrolled in this retrospective cohort study. Detailed baseline characteristics and information on disease status and treatment for each patient are presented in Table 1. AML with R/R status accounted for in 68% of cases (83 of 122), while secondary or therapy-related AML accounted for in 20.5% of cases (25 of 122). Decitabine (92.6%, 113/122) was the predominant combination agent, followed by azacitidine (4.9%, 6/122), and LDAC (2.5%, 3/122). A total of 340 cycles of VEN-based combination chemotherapy were administered to 122 patients, with a median of 2 cycles (range, 1–10). A total of 85.3% (104 of 122) of patients discontinued therapy: HSCT (44.2%, 46/104) was the most common cause, followed by no response (23.1%, 24/104), and death (22.1%, 23/104).

The overall response rates were calculated in terms of CR/CRi (45.9%, 56/122) and MLFS (9.0%, 11/122). The median cycle of response achievement was one (range from one to four), and the median follow-up period from the initiation of VEN-based combination chemotherapy was 180 days (IQR, 110–303). The overall mortality for all patients was 43.4% (53 of 122), 33.3% (13 of 39) for ND patients, and 48.2% (40 of 83) for those with R/R status (*p* = 0.178). All patients were administered anti-fungal agents at the point of initiating VEN-based chemotherapy: Among them, 88.5% (108 of 122) received antifungal prophylaxis and 11.5% (14 of 122) received empirical or targeted therapy. Among the patients who were administered antifungal prophylaxis, 98.1% (106 of 108) were treated with fluconazole, while two were treated with posaconazole. Voriconazole (46.1%, 6 of 13) was the predominant anti-fungal agent used for targeted therapy, followed by liposomal amphotericin B (30.8%, 4 of 13), and itraconazole (23.1%, 3 of 13), for targeted or empirical agents.

### 3.2. IFIs in VEN-Based Combination Chemotherapy

#### 3.2.1. Characteristics and Incidence

Twenty-two episodes (9 episodes (40.9%) after response achievement and 13 episodes (59.1%) without response) of IFIs were reported without recurrent episodes of IFIs in the same patients. The cumulative frequency of occurrence was 18% (22 of 122) and the incidence was 6.6 IFIs/100 cycles: 4.8 IFIs/100 cycles in de novo AML and 45.0 IFIs/100 cycles in secondary or therapy-related AML. Episodes occurred a median of two cycles (IQR, 1–2) and 55 days (IQR, 26–80) after the initiation of VEN-based combination chemotherapy. There were no different characteristics between early onset IFIs (within the second cycle of VEN-based combination chemotherapy) and late onset IFIs (occurring in the third cycle or more) (Appendix A). IFIs, including two proven and 20 probable cases, were identified and analyzed. There were also 11 possible IFIs cases that were not included in this study. *Aspergillus* was the only observed cause of IFIs, and the respiratory tract was the most common site of infection (95.5%, 21/22), followed by one case in the sinuses (4.5%, 1/22). Most patients (19/22, 86.4%) were diagnosed with IFIs when fluconazole was administered as prophylaxis, and only three cases (13.6%) of breakthrough IFIs developed in patients who were treated using mold active agents: one in a patient treated with posaconazole prophylaxis and two in patients empirically treated with itraconazole. Profound neutropenia (81.8%, 18/22) and prolonged neutropenia (86.4, 19/22) were present in most cases. Detailed patient characteristics according to IFIs are presented in Table 2.

#### 3.2.2. Risk Factors and Outcomes

The univariate logistic analysis indicated that IFIs occurred more frequently in patients with secondary or therapy-related AML than in those with de novo AML (odds ratio (OR), 3.635; 95% confidence interval (CI); 1.332–9.920, *p* = 0.011) and in patients treated with >20 mg of prednisolone for ≥2 weeks within 90 days of VEN-based combination chemotherapy (OR, 4.222; 95% CI, 1.033–17.260; *p* = 0.045). (Table 3A). In the multivariate analysis, only secondary or therapy-related AML was independently associated with IFIs (OR, 3.859; 95% CI, 1.344–11.048; *p* = 0.012).

The overall mortality rate was 43.4% (53 of 122): Mortality rate wase 63.6% (14 of 22) in patients with IFIs and 39.0% (39 of 100) in those without IFIs (OR, 2.737; CI, 1.051–7.128; *p* = 0.034). IFIs accounted for 17.0% (9 of 53) of all deaths. In the Kaplan-Meier survival analysis of the time to death from any cause at the end of the study period after the initiation of VEN-based combination chemotherapy, patients without IFIs exhibited a significantly better survival rate than patients who developed IFIs (Figure 1A).

### 3.3. BSIs in VEN-Based Combination Chemotherapy

#### 3.3.1. Characteristics and Incidence

We identified 39 episodes of BSIs in 35 patients (28.7% frequency of occurrence), and the incidence rate was 12.7 BSIs/100 cycles: Four patients had recurrent episodes of BSIs during VEN-based combination chemotherapy. The first episodes occurred a median of two cycles (IQR, 1–2) and 42 days (IQR, 19–67) after the initiation of therapy, and 12 (30.8%) episodes developed with empirical or targeted antibacterial agents. Overall, Gram-negative bacteria were more predominant than Gram-positive bacteria (61.5%, 24/39 vs. 38.5%, 15/39, respectively); however, Gram-positive bacteria were the leading pathogens in breakthrough BSIs (83.3%, 10/12 vs. 16.7%, 2/12, respectively). MDR organisms were confirmed in 14 episodes of BSIs (35.9%) which included nine episodes of Gram-positive organisms and five episodes of Gram-negative organisms (60%, 9/15 of Gram-positive organisms vs. 20.8%, 5/24 of Gram-negative organisms, OR; 5.412, 95% CI; 1.312–24.884, *p* = 0.013) (Appendix A). Mucosal injury or gastrointestinal origin (76.9%, 30/39) were the most common causes of BSIs, followed by primary bacteremia (12.8%, 5/39) and central line-associated BSI (10.3%, 4/39). Profound neutropenia (89.7%, 35/39) and prolonged neutropenia (84.6%, 33/39) were present in most cases. Detailed patient characteristics according to the first episode of BSIs are provided in Table 4.

#### 3.3.2. Risk Factors and Outcome

In the univariate analysis, a history of >20 mg of prednisolone treatment for ≥2 weeks within 90 days of VEN-based combination chemotherapy (OR, 5.793; 95% CI, 1.361–24.665; *p* = 0.017) was associated with a high incidence of BSIs. However, the type of AML or AML status at the initiation of VEN-based chemotherapy or responses did not affect the occurrence of BSIs (Table 3B). In the multivariate analysis, only the history of steroid administration was independently associated with BSI (OR, 7.474; 95% CI; 1.661–3.631, *p* = 0.008).

Overall mortality rates were 42.9% (15 of 35) in patients with BSIs and 43.7% (38 of 87) in patients without BSIs (OR, 0.967; CI, 0.438–2.136; *p* = 0.934). BSIs accounted for 9.4% (5 of 53) of all deaths. There was no significant difference or trend in overall survival in the analysis of BSIs (Figure 1B).

## 4. Discussion

This study aimed to investigate the incidence of IFIs/BSIs, their risk factors, and their impact on the outcomes of infectious complications in AML patients undergoing VEN-based combination chemotherapy and provide implications for the optimal selection of antimicrobial prophylaxis. Our results demonstrated that 22 patients developed IFIs (18%) during repeated chemotherapy, with an incidence of 6.6 IFIs/100 cycles. Remarkably, secondary, or therapy-related AML significantly increased the cumulative occurrence and incidence of IFIs. IFIs also resulted in worse outcomes in patients treated with VEN-based combination chemotherapy. These findings indicate the use of mold-active antifungal agents in a high-risk group such as secondary or therapy-related AML patients. BSIs were reported in 35 patients (28.7%) with an incidence of 12.7 BSIs/100 cycles. However, unlike patients who developed IFIs, the occurrence of BSIs was not related to the patient’s basal characteristics and overall survival. Rather, we identified that the concurrent use of antibacterial agents might cause BSIs by MDR pathogens.

Antimicrobial prophylaxis is recommended to prevent infectious complications, due to a prolonged and profound neutropenia when intensive chemotherapy is administered to patients with AML [12,13,23]. Although VEN-based combination chemotherapy has emerged as a promising option for AML treatment and is known to cause prolonged and profound neutropenia, there is no consistent consensus for antimicrobial prophylaxis in patients treated with VEN-based combination chemotherapy [24,25]. In studies of VEN monotherapy for chronic lymphocytic leukemia and early clinical studies of VEN-based combination chemotherapy for AML, the authors reported that cytopenia and tumor lysis syndrome were major side effects and that increases in severe infectious complications were not evident [26,27].

In this study, fluconazole prophylaxis was administered to 86.9% of the study population. IFIs were reported in 18% (22 of 122) of patients, most of whom (86.4%, 19/22) were diagnosed with IFIs while using fluconazole. This rate is higher than those reported in previous studies, which have noted that IFIs occur in up to 6.9–12.6% of patients during chemotherapy [14,16]. Rather, our results were similar to those reported for patients receiving salvage VEN-based combination chemotherapy (IFIs in up to 19.0% of patients) when in the relapsed state after transplantation [14,16,28]. These differences may be due to the selection of drugs for antifungal prophylaxis and the characteristics of the enrolled patients [14,16]. First, mold-active antifungal agents, such as posaconazole, voriconazole, and isavuconazole, were used in 50–100% of patients in previous studies, whereas they were administered in only 13.1% of patients in our study. Second, patients with R/R status—a known risk factor for IFIs-accounted for 68.0% of our population, which is higher than the rates of 48.8–54.0% reported in previous studies [14,16,28]. However, our results only indicated that IFIs tended to occur more frequently in patients with R/R status and non-responders, without statistical significance [14]. Rather, secondary or therapy-related AML patients were an identified independent predictor of IFIs. Notably, only 20.5% of our patients had secondary or therapy-related AML. This rate is lower than the 47–56.9% reported in previous studies [14,16]. Lastly, high dose VEN (600 mg/d) might affect the incidence of IFIs but statistical analysis was limited, as only three patients were administered high dose VEN. In this regard, antifungal prophylactic agents might have had a major effect on the higher incidence of IFIs than characteristics of the patients.

Azoles are widely used as prophylactic agents, and posaconazole shows superior efficacy when compared to fluconazole or itraconazole in patients with AML performing remission induction chemotherapy [11]. The results of this study did not show a difference of efficacy between antifungal agents, as almost 90% of this cohort patients were administered fluconazole uniformly. However, we demonstrated that secondary or therapy-related AML was an independent risk factor of IFIs, and had a three times higher incidence rate than de novo AML (4.8 IFIs/100 cycles in de novo AML and 15.0 IFIs/100 cycles in secondary or therapy-related AML). Therefore, fluconazole could be a viable option for low-risk patients, such as in newly diagnosed de novo AML without a history of steroid therapy. However, the use of mold-active antifungal agents is a generally profitable choice, and it should be considered, especially in those with secondary or therapy-related AML.

The occurrence of IFIs during VEN-based combination chemotherapy resulted in a high mortality and had a significant impact on survival, which was consistent with previous reports [11,29]. The causative organisms of IFIs were all *Aspergillus* species, all but one case, involving the sinuses, involved the respiratory tract, in accordance with previous findings [14,30]. Although one study reported the occurrence of breakthrough IFIs caused by mucormycosis, all three cases of breakthrough IFIs in our study (one in patient receiving using posaconazole and two in patients receiving itraconazole) were caused by *Aspergillus* [14]. These results also provide strong evidence for the use of mold-active antifungal prophylaxis being essential in risk defined groups. Furthermore, a critical review of treatment approaches to these high-risk patients is needed in order to improve their survival.

To the best of our knowledge, no studies have examined the BSIs in patients undergoing VEN-based combination chemotherapy. In this study, routine antibacterial prophylaxis was not administered during the study period. Thirty-nine episodes of BSIs occurred in 35 (28.9%) patients, four of whom experienced recurrent episodes. Recent studies reported that fluoroquinolone prophylaxis and the empirical use of broad-spectrum antimicrobial agents in patients with hematologic malignancies increase the frequency of infection caused by Gram-positive bacteria and their resistance rates [31,32]. In this study, Gram-negative bacteria prevailed overall. However, in breakthrough BSIs that occurred during the use of empirical or targeted antibacterial agents, Gram-positive bacteria were dominant, similar to findings observed in patients treated with prophylactic antibacterial agents at this study institute [33]. Moreover, Gram-positive bacteria had a significantly higher rate of MDR than Gram-negative bacteria. The risk of developing BSIs increased if there was a history of high-dose steroid use, as well as in patients with IFIs. In contrast, AML type, disease status at the initiation of VEN-based combination chemotherapy, risk group, and response did not affect BSIs’ incidence and, unlike IFIs, BSIs did not significantly affect the patient’s outcome. Furthermore, routine prophylactic antibacterial agents might lead to a breakthrough infection caused by MDR Gram-positive organisms. So further studies on antibacterial prophylaxis are needed and antibacterial prophylaxis should be considered carefully depending on the local epidemiology and strategy of antibacterial use.

This study had several limitations. First, this study was primarily designed as a single-centered, retrospective cohort study. This made it difficult to evaluate the causal relationship between covariates and the incidence of IFIs or BSIs. Moreover, a modest number of patients were enrolled, limiting the statistical power of the study. Second, various covariates could not be adjusted as the frequency of infectious complications was low. Lastly, socio-environmental factors affecting the development of IFIs were not collected and adjusted for retrospective study design. Despite these limitations, our study has several strengths: Our observations are reflective of the real-world data with a sizable number of homogenous patients treated with the same antimicrobial prophylaxis protocol at a single institution. Among the reported retrospective cohort studies, this study included the largest number of patients with AML. Second, we identified a high frequency of occurrence of IFIs and BSIs, even with a small cycle of chemotherapy, and assessed the risk factors and their outcomes. Although there are still unanswered questions regarding antimicrobial prophylaxis, these findings allow for a meaningful and robust assessment of IFIs and BSIs in patients receiving VEN-based combination chemotherapy.

## 5. Conclusions

This study demonstrated that the incidence of IFIs and BSIs during VEN-based combination chemotherapy is substantial. IFIs increased the mortality rate, and secondary and therapy-related AML was documented as an independent risk factor for an increased incidence of IFIs. Our findings also suggest that antifungal prophylaxis is available, and that mold-active antifungal agents should be a mandatory option for prophylaxis, especially in high-risk patients (e.g., secondary or therapy-related AML). However, BSIs did not affect patient outcomes, and no BSI risk groups could be identified. Antibacterial prophylaxis should be carefully considered as it may increase the risk of breakthrough infection by resistant pathogens.

## Figures and Tables

**Figure 1 cancers-13-06285-f001:**
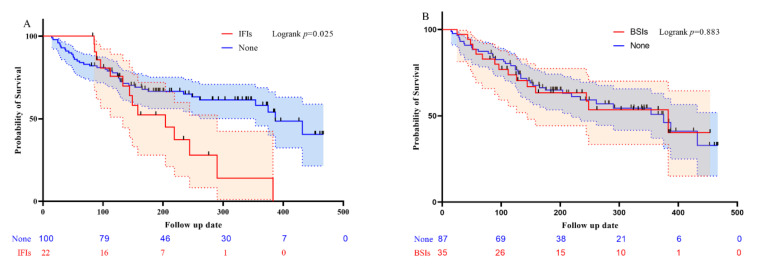
Overall survival for patients with AML treated with VEN-based combination chemotherapy. The red and blue shade bands represent approximated 95% confidence interval. (**A**) Overall survival according to IFIs. (**B**) Overall survival according to BSIs. BSIs: bloodstream infection; IFIs: invasive fungal infections.

**Table 1 cancers-13-06285-t001:** Baseline characteristics, prophylaxis, responses, and outcomes of study patients.

Variables	Total = 122 (Number, %)
Sex (male)	59 (48.4)
Age, years (median, IQR)	61 (47–70)
AML type at diagnosis	
De novo/MRC	97 (79.5)
Secondary	21 (17.2)
Therapy-related	4 (3.3)
AML status at initiation of VEN-based therapy	
Newly diagnosed	39 (32.0)
Refractory/relapsed	83 (68.0)
Prior treatment before VEN-based therapy	
Naïve	39 (32.0)
Intensive chemotherapy	38 (31.1)
Hypomethylating agents	9 (7.4)
HSCT	36 (29.5)
AML risk group	
Favorable	25 (20.5)
Moderate	46 (37.7)
Poor	51 (41.8)
Combination agents	
Decitabine	113 (92.6)
Azacitidine	6 (4.9)
Low-dose cytarabine	3 (2.5)
Overall response	
CR + CRi	56 (45.9)
MLFS	11 (9.0)
Non response	51 (41.8)
Not available	4 (3.3)
Completion of VEN-based therapy	104 (85.3)
HSCT	46 (44.2)
Non response	24 (23.1)
Death	23 (22.1)
Other reasons	11 (10.6)
Total cycle of VEN-based therapy (median, range)	2 (1–10)
Cycle of response achievement (median, range)	1 (1–4)
Antifungal agents	122 (100)
Empirical or targeted	14 (11.5)
Antifungal prophylaxis	108 (88.5)
Fluconazole	106 (98.1)
Posaconazole	2 (1.9)
Overall mortality	53 (43.4)

AML: acute myeloid leukemia; CR: complete recovery; CRi: complete recovery with incomplete hematologic recovery; HSCT: hematopoietic stem cell transplantation; IQR: interquartile range; MLFS: morphologic leukemia free state; MRC: myelodysplasia-related changes; VEN: venetoclax.

**Table 2 cancers-13-06285-t002:** Characteristics in patients according to invasive fungal infections.

Variables	Total = 122 (Number, %)	Without IFIs = 100 (Number, %)	IFIs = 22 (Number, %)	*p*
Sex (male)	46 (46.0)	13 (59.1)	59 (48.4)	0.381
Age, years (median, IQR)	61.0 (47.0–70.0)	58.5 (45.5–69.5)	62.0 (49.0–71.0)	0.401
AML type at diagnosis				0.057
De novo/MRC	97 (79.5)	84 (84.0)	13 (59.1)	
Secondary	21 (17.2)	14 (14.0)	7 (31.8)	
Therapy-related	4 (3.3)	2 (2.0)	2 (9.1)	
AML status at initiation of VEN-based therapy				0.788
Newly diagnosed	39 (32.0)	33 (33.0)	6 (27.3)	
Refractory/relapsed	83 (68.0)	67 (67.0)	16 (72.7)	
AML risk group				0.518
Favorable	25 (20.5)	19 (19.0)	6 (27.3)	
Moderate	46 (37.7)	37 (37.0)	9 (40.9)	
Poor	51 (41.8)	44 (44.0)	7 (31.8)	
Combination agents				0.463
Decitabine	113 (92.6)	94 (94.0)	19 (86.4)	
Azacitidine	6 (4.9)	4 (4.0)	2 (9.1)	
Low-dose cytarabine	3 (2.5)	2 (2.0)	1 (4.5)	
Overall response				0.275
CR + CRi	56 (45.9)	49 (49.0)	7 (31.8)	
MLFS	11 (9.0)	7 (7.0)	4 (18.2)	
Non response	51 (41.8)	41 (41.0)	10 (45.5)	
Not available	4 (3.3)	3 (3.0)	1 (4.5)	
Antifungal agents at development of IFIs				1.000
Empirical or targeted	14 (11.6)	12 (12.1)	2 (9.1)	
Antifungal prophylaxis	108 (88.4)	88 (87.9)	20 (90.9)	
Fluconazole	106 (97.2)	87 (97.7)	19 (95.0)	
Posaconazole	2 (1.9)	1 (1.1)	1 (5.0)	
Type of antifungal agents at development of IFIs				1.000
Fluconazole	106 (86.9)	87 (87.0)	19 (86.4)	
Mold active antifungal agents	16 (13.1)	13 (13.0)	3 (13.6)	
History of IFIs within 3 months	16 (13.1)	13 (13.0)	3 (13.6)	1.000
Steroid use before IFIs developed	9 (7.4)	5 (5.0)	4 (18.2)	0.091
Overall mortality	53 (43.4)	39 (39.0)	14 (63.6)	0.061

AML: acute myeloid leukemia; CR: complete recovery; CRi: complete recovery with incomplete hematologic recovery; IFIs: invasive fungal infections; IQR: interquartile range; MLFS: morphologic leukemia free state; MRC: myelodysplasia-related changes; VEN: venetoclax.

**Table 3 cancers-13-06285-t003:** (**A**) Risk factors for IFIs during VEN-based combination chemotherapy in univariate analysis and multivariate analysis (**B**) risk factors for BSIs.

**A. Variables**	**Univariate Analysis**		**Multivariate Analysis**	
	**OR**	**95% CI**	**OR**	**95% CI**
Sex (male vs. female)	1.695	0.665–4.325		
Age, years (over 60 years)	1.750	0.675–4.539		
AML type at diagnosis				
De novo/MRC	Reference		Reference	
Secondary/Therapy-related	3.635	1.332–9.920	3.859	1.344–11.048
AML status at initiation of VEN-based therapy				
Newly diagnosed	Reference		Reference	
Refractory/relapsed	1.313	0.470–3.667	1.228	0.405–3.722
Prior treatment before VEN-based therapy				
Naïve	Reference			
Intensive chemotherapy	1.467	0.456–4.717		
Hypomethylating agents	1.572	0.261–9.470		
HSCT	1.100	0.320–3.782		
AML risk group				
Favorable	Reference			
Moderate	0.770	0.239–2.486		
Poor	0.504	0.149–1.700		
Combination agents				
Decitabine	Reference			
Azacitidine	2.474	0.422–14.487		
Low-dose cytarabine	2.474	0.213–28.682		
Overall response				
Response group	Reference			
Non response group	1.295	0.503–3.340		
Overall antifungal agents				
Fluconazole	Reference			
Mold active antifungal agents	1.057	0.274–4.076		
Steroid use before IFIs developed	4.222	1.033–17.260	4.266	0.941–19.331
**B. Variables**	**Univariate Analysis**		**Multivariate Analysis**	
	**OR**	**95% CI**	**OR**	**95% CI**
Sex (male)	1.012	0.462–2.218		
Age, years (over 60 years)	1.303	0.591–2.872	1.297	0.463–3.631
AML type at diagnosis				
De novo/MRC	Reference		Reference	
Secondary/Therapy-related	1.22	0.472–3.156	1.149	0.427–3.090
AML status at initiation of VEN-based therapy				
Newly diagnosed	Reference		Reference	
Refractory/relapsed	0.721	0.316–1.646	0.652	0.226–1.878
Prior treatment before VEN-based therapy				
Naïve	Reference			
Intensive chemotherapy	0.621	0.228–1.689		
Hypomethylating agents	0.572	0.104–3.149		
HSCT	0.880	0.333–2.328		
AML risk group				
Favorable	Reference			
Moderate	0.591	0.212–1.648		
Poor	0.461	0.165–1.291		
Combination agents				
Decitabine	Reference			
Azacitidine	NA	NA		
Low-dose cytarabine	4.848	0.425–55.320		
Overall response				
Response group	Reference			
Non response group	1.029	0.463–2.286		
Steroid use before BSIs developed	5.793	1.361–24.665	7.474	1.661–33.622

AML: acute myeloid leukemia; BSIs: bloodstream infections; CI: confidence interval; HSCT: hematopoietic stem cell transplantation; IFIs: invasive fungal infections; MRC: myelodysplasia-related changes; OR: odds ratio; VEN: venetoclax.

**Table 4 cancers-13-06285-t004:** Characteristics in patients according to bloodstream infections.

Variables	Total = 122 (Number, %)	Without BSIs = 87 (Number, %)	BSIs = 35 (Number, %)	*p*
Sex (male)	59 (48.4)	42 (48.3)	17 (48.6)	1.000
Age, years (median, IQR)	61.0 (47.0–70.0)	60.0 (46.0–68.0)	62.0 (47.5–71.5)	0.274
AML type at diagnosis				0.011
De novo/MRC	97 (79.5)	70 (80.4)	27 (77.1)	
Secondary	21 (17.2)	15 (17.2)	6 (17.1)	
Therapy-related	4 (3.3)	2 (2.3)	2 (5.7)	
AML status at initiation of VEN-based therapy				0.573
Newly diagnosed	39 (32.0)	26 (29.9)	13 (37.1)	
Refractory/relapsed	83 (68.0)	61 (70.1)	22 (62.9)	
AML risk group				0.328
Favorable	25 (20.5)	15 (17.2)	10 (28.6)	
Moderate	46 (37.7)	33 (37.9)	13 (37.1)	
Poor	51 (41.8)	39 (44.8)	12 (34.3)	
Combination agents				0.103
Decitabine	113 (92.6)	80 (92.0)	33 (94.3)	
Azacitidine	6 (4.9)	6 (6.9)	0 (0.0)	
Low-dose cytarabine	3 (2.5)	1 (1.1)	2 (5.7)	
Overall response				0.133
CR + CRi	56 (45.9)	42 (48.3)	14 (40.0)	
MLFS	11 (9.0)	5 (5.7)	6 (17.1)	
Non response	51 (41.8)	36 (41.4)	15 (42.9)	
Not available	4 (3.3)	4 (4.6)	0 (0.0)	
Steroid use before BSI developed	9 (7.4)	3 (3.4)	6 (17.1)	0.025
Overall mortality	53 (43.4)	38 (43.7)	15 (42.9)	1.000

AML: acute myeloid leukemia; BSIs: bloodstream infections; CR: complete recovery; CRi: complete recovery with incomplete hematologic recovery; IQR: interquartile range; MLFS: morphologic leukemia free state; MRC: myelodysplasia-related changes; VEN: venetoclax.

## Data Availability

Original data will be available without any restrictions with digital object identifier after the publication is decided.

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
