# Peer review of "Infections of Venetoclax-Based Chemotherapy in Acute Myeloid Leukemia: Rationale for Proper Antimicrobial Prophylaxis"

_cancers, 2021, doi:10.3390/cancers13246285_

Round 1
Reviewer 1 Report
This is only an observational study coming from a single center patient population. For this reason you can only describe the situation. You cannot draw definitive conclusions or recommendations or "provide evidences !"! Please avoid sentences like "provide optimal evidence" (lines 15-16), "strongly " recommended (line 21), "strong evidence" (line 37)
Delete "provide evidence for optimal antimicrobial prophylaxis in VEN-based combination chemotherapy " (line 71).
The DISCUSSION IS TOO LONG AND REPETITIVE-PLEASE REDUCE IT BY 50%
REMOVE FROM THE DISCUSSION "Few studies have provided evidence regarding the effects of antimicrobial prophylaxis in patients with AML receiving VEN-based combination chemotherapy (line 226)". This is not an useful and appropriate introduction - your study is observational and retrospective. It is unable to provide any EVIDENCE !.
Please briefly comment in the DISCUSSION SECTION that you are using a high dose of venetoclax (600 mg/d) and you have 68% of case with R/R AML - this could affect the high infectious rate.
Remove "STRONGLY SUPPORT (line 235)" and replace with suggest/indicate etc
Remove "IS MANDATORY" and replace with "is advisable" (line 335)
Author Response
Comments and Suggestion 1: This is only an observational study coming from a single center patient population. For this reason you can only describe the situation. You cannot draw definitive conclusions or recommendations or "provide evidences !" Please avoid sentences like "provide optimal evidence" (lines 15-16), "strongly " recommended (line 21), "strong evidence" (line 37)
Response: We strongly agree that the limitation of this study is its retrospective observational design. As you mentioned, we revised and deleted definitive phrases. Please see lines 16, 21 and 36.
Comments and Suggestion 2: Delete "provide evidence for optimal antimicrobial prophylaxis in VEN-based combination chemotherapy " (line 71).
Response: Thank you for your comments. As you mentioned, this study has limitations in drawing definitive conclusions. To clarify the text, we revised and deleted this sentence. Please see lines 71-72.
Comments and Suggestion 3: The DISCUSSION IS TOO LONG AND REPETITIVE-PLEASE REDUCE IT BY 50%
Response: We appreciated your comment and suggestion and we are agree with your opinion. We have corrected and clarify the discussion section by reducing repetitive phrases. Please see discussion section lines 227-228, 251-254, 255-257, 273-274, 300-301, 314-317. However, we humbly request you to kindly let us remain the rest of discussion although we could not reduce the discussion section by 50% as you comment. All authors cross-checked the discussion and the remained text is considered essential for the completeness and in-depth understanding of our research.
Comments and Suggestion 4: REMOVE FROM THE DISCUSSION "Few studies have provided evidence regarding the effects of antimicrobial prophylaxis in patients with AML receiving VEN-based combination chemotherapy (line 226)". This is not an useful and appropriate introduction - your study is observational and retrospective. It is unable to provide any EVIDENCE !.
Response: We really appreciated your detailed comment. Above sentence is not appropriate as this study and previous studies could not provide optimal evidence for antimicrobial prophylaxis. We deleted this sentence from the first paragraph of discussion. Please see lines 227-228.
Comments and Suggestion 5: Please briefly comment in the DISCUSSION SECTION that you are using a high dose of venetoclax (600 mg/d) and you have 68% of case with R/R AML - this could affect the high infectious rate.
Response 1: We completely agree with your comments and suggestion. In the previous study, R/R AML was identified as risk factors for developing invasive fungal infections. Therefore, high proportion of R/R status in this study might affect the notable incidence of IFIs in this study. We mentioned this in lines 267-271.
Response 2: Thank you for your detailed comment. In this study, 600mg/d of venetoclax was only administered only in 3 patients with low-dose cytarabine. As you comment, venetoclax dose may have an effect on IFI incidence, but no similar report was found in previous studies. Furthermore, statistical analysis was limited as only 3 patients were administered high dose VEN in this study. However, as you suggestion, it could be the important aspect to consider, we revised manuscript. Please, see lines 275-277.
Comments and Suggestion 6: Remove "STRONGLY SUPPORT (line 235)" and replace with suggest/indicate etc.
Response: ‘Strongly support’ was revised to ‘indicate’. Please see line 236.
Comments and Suggestion 7: Remove "IS MANDATORY" and replace with "is advisable" (line 335)
Response: Thank you for your detailed comment. ‘Strongly support’ was revised to ‘indicate’. Please see line 341.
From authors to reviewer: During the revision, it was confirmed that there was an important typing error in the discussion part of manuscript. The incidence of secondary or therapy related AML was 15.0 IFIs/100 cycles not 45.0 IFIs/100cycles. This mistake was caused by misreading the number 1 as the number 4. It was reconfirmed that this mistake was a simple typing error and had no effect on the analysis of the results of the research. Please, see lines 284-285. Again we really apologize for the mistake.
Reviewer 2 Report
The authors describe a retrospective experience on antimicrobial prophylaxis in AML patients treated on venetoclax-based regimens. The topic is of interest, however limited by the retrospective design of the study.
Following my comments:
- The presentation of data needs some clarifications: line 120 AML R/R 68% of cases is not clear as compared to table 1 in which the authors state that DeNovo AML account for 79,5% of cases. Data need explanation,
- Table 2 contains a duplication of information reported in table 1
- Patients can be stratified according to risk profile for fungal infections (based on personal history, job etc). The author should comment on this.
- table 4: is the legend correct?
- line 254: the sentence is not clear
Author Response
We would like to thank the editor and reviewers for the invaluable comments that helped a great deal in improving our original manuscript. Please find attached our revised paper and below a summary of how we responded to the comments.
Comments and Suggestion 1: The presentation of data needs some clarifications: line 120 AML R/R 68% of cases is not clear as compared to table 1 in which the authors state that De Novo AML account for 79,5% of cases. Data need explanation
Response: Thank you for your detailed comments. We classified de novo or secondary/therapy-related AML at the first time of AML diagnosis of each patient. On the other hand, newly diagnosed or R/R AML was classified according to the AML status at the time of VEN-based combination chemotherapy. Therefore, there may be differences between them. For example, even with the same de novo AML patient at the initial diagnosis, if VEN-based chemotherapy was used as frontline-therapy, it corresponds to newly diagnosed, but if VEN-based chemotherapy was used for salvage therapy after relapse, it was included in R/R status AML. Please see Table 1.
Comments and Suggestion 2: Table 2 contains a duplication of information reported in table 1
Response: We strongly agree that Table 2 contained duplicate information reported in Table 1. However, please consider that the purpose of the tables is different, so even if there is overlapping information, it is necessary to separate the tables. Table 1 contains the general baseline characteristics of study population and allows readers to check the characteristics of the target patient group at a glance. Rather, Table 2 and Table 4 classify the characteristics according to the occurrence of invasive fungal infections and bloodstream infections so that the differences between each group can be quickly compared before statistical analysis. Moving table2 and table 4 to supplementary materials could be a one way to solve it, but considering that many readers do not even look at supplementary materials, we believe it would be good to keep tables in the manuscript for different purposes.
Comments and Suggestion 3: Patients can be stratified according to risk profile for fungal infections (based on personal history, job etc. The author should comment on this.
Response: We really appreciate your helpful comment and suggestion. The incidence of IFIs is associated with various factors, especially socio-environmental factors as you mentioned. Therefore, it is important to stratify or adjust for these factors to confirm the occurrence of IFIs. However, because this was a retrospective study, we could not collect these various factors through chart review. We included information in line with this in the limitations section. Although socio-environmental factors were not stratified, this study results were analyzed after adjusting for various factors that might affect the development of IFIs. Please see lines 327-328.
Comments and Suggestion 4: table 4: is the legend correct?
Response: Thank you for your kind and detailed comment. There was an error while writing the manuscript. We sincerely apologize, and thank you for the correction. Please, see Table 4.
Comments and Suggestion 5: line 254: the sentence is not clear
Response: We checked the error caused by using the verb twice. This sentence was deleted as this phrase is repetitive in discussion section. Please see lines 255-257.
From authors to reviewer: During the revision, it was confirmed that there was an important typing error in the discussion part of manuscript. The incidence of secondary or therapy related AML was 15.0 IFIs/100 cycles not 45.0 IFIs/100cycles. This mistake was caused by misreading the number 1 as the number 4. It was reconfirmed that this mistake was a simple typing error and had no effect on the analysis of the results of the research. Please, see lines 284-285. Again we really apologize for the mistake.

Reviewer 3 Report
The subject of this manuscript, i.e., the risk of serious infections in AML patients treated with venetoclax in combination with hypomethylating agents or conventional cytotoxic drugs, is relevant and important. The results of this retrospective analysis from a single, high-volume center indicate the need for systemic antibacterial and mold-active antifungal prophylaxis, similar to conventional myelosuppressive AML induction chemotherapy.
A few remarks:
There are already informative and in part detailed data on this subject in the literature: Guo et al, Hematology 2020 (review), and DiNardo et al in their NEJM paper, where fever and infections are pretty precisely described.
An incidence of 18% invasive Aspergillus infections in AML patients (receiving fluconazole prophylaxis) is alarming. Typically, in placebo arms of randomized studies on mold-active azole prophylaxis, this figure is around 7-8%. In the analysis of risk factors provided in Table 2, neither the AML status (newly diagnosed vs relapsed/refractory) nor the achievement of a CR/CRi have been predictive, so it must be discussed that such a high incidence of invasive aspergillosis is not acceptable, as is the mortality rate of 63.6% among these patients. The conclusion should not be that mold-active antifungal prophylaxis should be considered, but rather that it is mandatory, as well as the critical review of treatment approaches to these patients in order to significantly improve their survival.
A bloodstream infection rate of 28.7% is also a critical number. Many of these BSI could have been prevented by quinolone prophylaxis, i.e., by cipro- or levofloxacin, because the majority of the BSI were caused by gram-negative bacteria. A consideration of spreading antimicrobial resistance is always justified, but in light of a 42.9% mortality rate in patients with BSI in their cohort, the authors must reconsider their reluctance. Meta-analyses (most recently Egan G et al, Cancer Med 2019;8:4536-46) and evidence-based guidelines (e.g., Mikulska M et al, J Infect 2018;76:20-37) should be referred to.
It is somewhat irritating that the authors describe universal fluconazole prophylaxis (which, by the way, is not backed-up by any guideline on antifungal prophylaxis in AML patients undergoing myelosuppressive treatment), while in their more detailed description of patient characteristics, they mention a small number of posaconazole recipients.
There are minimal typos: page 2, line 78, should read "transplantations" rather than "transplantation". More importantly, the title of Table 4 should be changed to "blood stream infections" rather than "invasive fungal infections".
Author Response
We would like to thank the editor and reviewers for the invaluable comments that helped a great deal in improving our original manuscript. Please find attached our revised paper and below a summary of how we responded to the comments.
Comments and Suggestion 1: An incidence of 18% invasive Aspergillus infections in AML patients (receiving fluconazole prophylaxis) is alarming. Typically, in placebo arms of randomized studies on mold-active azole prophylaxis, this figure is around 7-8%. In the analysis of risk factors provided in Table 2, neither the AML status (newly diagnosed vs relapsed/refractory) nor the achievement of a CR/CRi have been predictive, so it must be discussed that such a high incidence of invasive aspergillosis is not acceptable, as is the mortality rate of 63.6% among these patients. The conclusion should not be that mold-active antifungal prophylaxis should be considered, but rather that it is mandatory, as well as the critical review of treatment approaches to these patients in order to significantly improve their survival.
Response 1: We really appreciate your reviewing and comments. We strongly agree that incidence of IFIs in this study is a big challenge to treat the AML patients with VEN-based combination therapy. As you mentioned, we failed to test the statistical significance of the impact of AML status or responses on the occurrence of IFI. However, similar to previous studies, we also noticed a trend for IFIs to occur frequently in patients with R/R status (4.1 IFIs/100 cycles in newly diagnosed vs 8.6 IFIs/100 cycles in R/R status, not included in main manuscript text). In this study, the proportion of patients with R/R status was higher than in previous studies; it was described that the incidence might be high for this reason (please see lines 267-269). Notably, only identified risk group (secondary or therapy-related AML) was about 20%, which was lower than that in previous studies. In this regard, we postulate that fluconazole prophylaxis may have been the most important cause of its high incidence (please see lines 265-278).
Response 2: We also agree with your comment and suggestion about mold-active antifungal prophylaxis being mandatory. Due to the limitations of the retrospective study, we tried to refrain from assertive expressions as much as possible, but we revised the sentences to make them clearer. Please see lines 298-299 and 342-343.
Comments and Suggestion 2: A bloodstream infection rate of 28.7% is also a critical number. Many of these BSI could have been prevented by quinolone prophylaxis, i.e., by cipro- or levofloxacin, because the majority of the BSI were caused by gram-negative bacteria. A consideration of spreading antimicrobial resistance is always justified, but in light of a 42.9% mortality rate in patients with BSI in their cohort, the authors must reconsider their reluctance. Meta-analyses (most recently Egan G et al, Cancer Med 2019;8:4536-46) and evidence-based guidelines (e.g., Mikulska M et al, J Infect 2018;76:20-37) should be referred to.
Response: Thank you very much for your helpful comment and suggestion. All authors rigorously discussed the use of prophylactic antibacterial agents through the study results. As we already know, fluoroquinolone prophylaxis is an effective way to reduce the incidence of BSI. However, it could not reduce the mortality, rather it might be a risk factor for breakthrough BSI with MDR pathogens. In this study, breakthrough infection was confirmed to be more likely due to resistant bacteria and we also found that the mortality was not associated with the antibacterial use. That is the reason why current guidelines recommend prophylactic antibacterial agents tailored to local and institutional circumstances. In this study institute, ESBL producing and quinolone resistance rate of E. coli is 24% on average, and Klebsiella has a high resistance rate of over 40% (See below Figure 1, not published, not shown in the main manuscript. Provided by infection control team in this study institute). What is more worrisome is that the frequency of CRE is increasing, and the situation is similar not only in research institutes but also in Korea (See below Figure 2, not published, not shown in the main manuscript. Provided by infection control team in this study institute). In this regard, we now strictly restrain the use of fluoroquinolone in this institute and may need further evidence for wide use of antibacterial prophylaxis. However, per your comment, institutions with low resistance rates can expect BSIs reduction through FQ use. We revised and clarified this in the manuscript based on your comment and suggestion. Please see lines 319-321.
Figure 1. Rate of quinolone resistance of E. coli and K. pneumonia (At Seoul St. Mary’s hospital)
Figure 2. Incidence of CPE (At Seoul St. Mary’s hospital)
Comments and Suggestion 3: It is somewhat irritating that the authors describe universal fluconazole prophylaxis (which, by the way, is not backed-up by any guideline on antifungal prophylaxis in AML patients undergoing myelosuppressive treatment), while in their more detailed description of patient characteristics, they mention a small number of posaconazole recipients.
Response: We completely agree with your comment and consideration. When we developed a strategy for antimicrobial prophylaxis, there was a concern about whether it would be sufficient with fluconazole. That is the most important reason why we performed this study. There were two main reasons. First, as we described in the manuscript, VEN based combination chemotherapy in RCTs and earlier studies was considered to not significantly increase the risk of severe infection in patients with AML. Until now, there is no consensus on antifungal prophylaxis with a high level of evidence. Second (we did not mention in the manuscript text), Korea has a medical system that is operated under unified public health insurance. Therefore, there is a critical limitation i.e., many medical decisions are made within the scope of public insurance. Anti-mold active agents such as posaconazole are not approved except for intensive chemotherapy for AML, and unfortunately, VEN-based combination chemotherapy has not been categorized as an intensive chemotherapy for AML. This is why we used ‘universal fluconazole prophylaxis’. Based on our research, we are working on anti-mold agents for prophylactic use in VEN-based chemotherapy.
Comments and Suggestion 4: There are minimal typos: page 2, line 78, should read "transplantations" rather than "transplantation". More importantly, the title of Table 4 should be changed to "blood stream infections" rather than "invasive fungal infections".
Response: The manuscript was revised more clearly per your comment. Please see line 78. There was an error while writing the manuscript. We sincerely apologize and thank you for the correction. Please see Table 4.
From authors to reviewer: During the revision, it was confirmed that there was an important typing error in the discussion part of manuscript. The incidence of secondary or therapy related AML was 15.0 IFIs/100 cycles not 45.0 IFIs/100cycles. This mistake was caused by misreading the number 1 as the number 4. It was reconfirmed that this mistake was a simple typing error and had no effect on the analysis of the results of the research. Please, see lines 284-285. Again we really apologize for the mistake.

Round 2
Reviewer 3 Report
All revisions properly done.